# Prevalence and Correlates of Metabolic Syndrome and Its Components in Chinese Children and Adolescents Aged 7–17: The China National Nutrition and Health Survey of Children and Lactating Mothers from 2016–2017

**DOI:** 10.3390/nu14163348

**Published:** 2022-08-16

**Authors:** Jia Shi, Li He, Dongmei Yu, Lahong Ju, Qiya Guo, Wei Piao, Xiaoli Xu, Liyun Zhao, Xiaolin Yuan, Qiuye Cao, Hongyun Fang

**Affiliations:** National Institute for Nutrition and Health, Chinese Center for Disease Control and Prevention, Beijing 100050, China

**Keywords:** children and adolescents, metabolic syndrome, epidemiology, China

## Abstract

This descriptive study aimed to determine the prevalence of metabolic syndrome (MetS) and its components among Chinese children and adolescents aged 7–17 from 2016–2017 according to the Cook’s criteria modified for age on the basis of the National Cholesterol Education Program-Adult Treatment Panel III (NCEP-ATP III) and to evaluate the associations between the factors of interest (especially vitamin A, vitamin D and hyperuricemia) of MetS and its components, using data from the China National Nutrition and Health Survey of Children and Lactating Mothers from 2016–2017. A total of 54,269 school-aged children and adolescents were ultimately included in this study. Anthropometric measurements and laboratory examinations of the subjects and their relevant information were also collected. A multivariate logistic regression analysis model was applied to analyze the relationships between relevant factors associated with MetS and its components. In the present study, the prevalence of MetS in children and adolescents was 5.98%. Among the five components of MetS, elevated blood pressure (BP) and abdominal obesity were the most prevalent (39.52% and 17.30%), and 58.36% of the subjects had at least one of these components. In the multivariate logistic regression, an overweight condition, obesity and hyperuricemia were positively correlated with the incidence of MetS and all five components. There was also a positive association observed between vitamin A and the risk of MetS and some components of MetS (abdominal obesity and high triglycerides (TG)) and vitamin A was negatively associated with the risk of low high-density lipoprotein cholesterol (HDL-C). Subjects with vitamin D inadequacy had a higher risk of MetS (OR = 1.364, 95%CI: 1.240–1.500) and four of its components, excepting elevated FBG (fast blood glucose). Vitamin D deficiency was positively associated with MetS (OR = 1.646, 95%CI: 1.468–1.845) and all five of its components. Well-designed, large-scale prospective studies are also needed in the future.

## 1. Introduction

Metabolic syndrome (MetS) has become one of the major public health challenges worldwide and is characterized by the combination of abdominal obesity, elevated blood pressure (BP), high triglycerides (TG), elevated fast blood glucose (FBG) and low high-density lipoprotein cholesterol (HDL-C) [1]. The critical etiology of MetS are obesity and insulin resistance (IR), both of which can lead to glucose intolerance and dysglycemia [2]. The importance of diagnosing MetS lies in the fact that it helps to identify individuals at a high risk of cardiovascular diseases (CVDs) and type-2 diabetes [3]. Studies have established that several factors of MetS, particularly obesity, can track into adulthood [4,5] and predict metabolic syndrome in later life [6]. Therefore, the early identification of risk factors and MetS in children and adolescents is very valuable in targeting efforts for chronic disease prevention in adults.

Uria acid is a final product of purine nucleotides and is formed by the liver and mainly excreted by the kidneys in human metabolism [7]. Hyperuricemia can occur as a result of the overproduction or underexcretion of serum uric acid, but the relevant pathophysiological mechanism has not yet been fully elucidated [8]. One possible explanation is that this is a consequence of IR, and the other is that hyperuricemia is induced by excessive fructose intake because of an unhealthy diet [9]. Although epidemiologic studies have investigated the association of serum uric acid or hyperuricemia with MetS [2,9,10,11], little is known about its association in young populations in China, among which there is also a high prevalence of hyperuricemia and MetS.

Vitamin A is a lipid-soluble vitamin which plays important roles in the vision, the immune system and cell protection [12], and it has been observed that it also possesses anti-inflammatory properties. In particular, the carotenes, a kind of pro-vitamin A, have been reported to exhibit an antioxidant action [13,14]. Although the pathophysiology is not fully understood, numerous epidemiological studies have shown a significant inverse association between serum vitamin A and the incidence of MetS, mainly in adults [12,15]. However, the association of vitamin A with MetS and its components among subjects younger than 18 years old is still unclear in China. Therefore, we divided serum vitamin A concentration into three levels to investigate the possible associations.

Similarly, serum vitamin D is another fat-soluble molecule, mainly synthesized in the skin following exposure to UV-B radiation from sunlight or from the diet [16]. The functions of vitamin D include the promotion of bone mineralization, the inhibition of cellular proliferation and the induction of insulin production [17]. In recent years, the vitamin D insufficiency (≤20 ng/mL) has been described as a worldwide epidemic [18]. Many studies indicated that there is an inverse relationship between vitamin D insufficiency and the presence of MetS and its components [19,20,21,22]. However, the relationship remains controversial because no conclusive scientific evidence is available [23]. In the present study, we also divided vitamin D into three statuses according its serum concentrations to determine the associations.

In this study, data from the China National Nutrition and Health Survey of Children and Lactating Mothers from 2016–2017 were used to calculate the prevalence of MetS and its components and explore its possible correlates in order to provide evidence for the prevention and control of the disease. We also compared the corresponding indicators between subjects with and without MetS and its components to identify differences.

## 2. Materials and Methods

### 2.1. Study Design and Participants

Data were obtained from the China National Nutrition and Health Survey of Children and Lactating Mothers from 2016–2017. A complex, multistage, stratified cluster random sampling method was carried out to create a representative sample of 31 provinces, autonomous regions and municipalities. The first, including a total of 275 survey sites were, randomly selected. The second, including two townships/subdistricts, were randomly picked from every survey site. The third, two communities/villages were randomly selected from each township/subdistrict. At least 280 children and adolescents aged 6–18 were selected from schools in each survey site, with an equal number of males and females. In this study, after excluding participants who were unable to provide important information, such as total anthropometric measurement data and blood laboratory tests information, and participants who had unreasonable data, a total of 54,269 participants aged 7–17 were included.

### 2.2. Data Collection and Measurements

The data collection and measurements were conducted by the trained investigators. Basic demographic information, such as age, sex, living area and other information, was collected using the standardized questionnaires provided by the China Center for Disease Control and Prevention (the China CDC) project group.

The anthropometric measurements included height, weight, waist circumference (WC), systolic blood pressure (SBP), diastolic blood pressure (DBP) of the participants. Height and weight were measured without shoes or coats on. Height was measured to the nearest 0.1 cm with a stadiometer (TZG) while the participants were standing in an upright position. Weight was measured to the nearest 0.1 kg using an electric scale (G&G tc-200 k, Shanghai, China). BMI was calculated by dividing the body weight (kg) by the square of the height (m). The WC measurement was performed at the midpoint between the bottom of the rib cage and the top of the iliac crest twice using a soft tape. After guaranteeing that the error between the two measurements was less than 2 cm, the investigators then recorded the average value. After resting quietly for 5 min, the SBP and DBP were measured three times at one-minute intervals by an electronic sphygmomanometer (Omron HBP 1300, Tokyo, Japan) with an accuracy to 1 mmHg, and the mean of the SBP and DBP was recorded for further analysis. The measurements strictly followed the relevant standard protocols [24].

Laboratory examinations: 6 mL fasting venous blood samples of participants were obtained in the morning and used for the testing indicators. Triglyceride (TG) and high-density lipoprotein cholesterol (HDL-C) were tested using enzyme colorimetry (Roche Cobas C701 automatic biochemical analyzer series). Fast blood glucose (FBG) was measured using the glucokinase method (Roche P800 automatic biochemical analyzer). Vitamin A was tested by high-performance liquid chromatography (HPLC). Vitamin D was tested by high-performance liquid chromatography-mass spectrometry (HPLC-MS/MS). Serum uric acid was measured using colorimetry (Roche C702 automagical analyzer).

### 2.3. Ethics Approval and Consent to Participate

This study was approved by the ethical committee of the National Institute for Nutrition and Health of Chinese Center for Disease Control and Prevention, and the ethical approval number was 201614. All participants gave signed informed consent before the study, provided either by themselves or their guardians.

### 2.4. Diagnostic Criteria and Definitions

The living area was categorized into urban and rural areas. The nutritional status (normal, overweight and obese) of the participants was identified by the sex- and age-specific BMI cut-off values for Chinese school-age children and adolescents [25,26]. To assess the associations between MetS and age, we divided all the subjects into three age groups (7–10, 11–13 and 14–17 for females; 7–11, 12–14 and 15–17 for boys) to reflect the prepubertal, pubertal and post-pubertal stages respectively, according to the Chinese classifications [27].

Given that the Cook’s criteria [28] modified for age on the basis of the National Cholesterol Education Program-Adult Treatment Panel III (NCEP-ATP III) is sensitive in its identification of the risk factors of MetS [29] and shows moderate agreement that is appropriate to application to Chinese children and adolescents, we selected Cook’s criteria for defining MetS in this study.

According to the Cook’s criteria, participants meeting at least 3 of the following 5 criteria were defined as having the MetS:(1)Abdominal obesity: WC ≥ age- and sex-specific 90th percentile, determined by the cut-off points for Chinese children and adolescents [30];(2)Elevated blood pressure: SBP or DBP ≥ 90th percentile for age, sex and height [31];(3)High triglycerides: serum TG ≥ 1.24 mmol/L;(4)Low HDL-C: HDL-C ≤ 1.03 mmol/L;(5)Elevated fast blood glucose: FBG ≥ 6.1 mmol/L.

Hyperuricemia was defined in this study according to the threshold of the serum uric acid value ≥357 µmol/L, in line with previous studies [27].

Vitamin A sufficiency, inadequacy and deficiency were defined according to serum retinol levels of more than 0.3 µg/mL and 0.2–0.3 µg/mL, and less than 0.2 µg/mL, respectively [32]. Vitamin D sufficiency, inadequacy and deficiency were defined as serum 25(OH)D levels of more than 20 ng/mL and 12–20 ng/mL, and less than 12 ng/mL, respectively [33].

### 2.5. Statistical Analysis

All analyses were conducted with SAS software (v.9.4, SAS Institute Inc., Cary, NC, USA). The continuous data were displayed as the median (interquartile range, IQR) in view of non-normality. The categorical data were reported as numbers and percentages. The continuous variables were compared using the Wilcoxon rank-sum test between subgroups. The chi-square test was used to compare the distributions of different characteristics between subgroups. The Cochran–Armitage trend test was used to test the increasing/decreasing trends between subgroups. The prevalence of MetS, its risk factors and its components were calculated. Multivariate logistic regression analysis was used to explore the related factors of MetS and its components. The odds ratios (ORs) and 95% confidence intervals (CIs) were presented. Two-sided *p* values < 0.05 were considered to be statistically significant.

## 3. Results

### 3.1. Basic Characteristics of the Research Population

A total of 54,269 children and adolescents aged 7–17 were included in this study. The basic demographic, anthropometric and clinical characteristics of the research population are presented in Table 1. Among the study participants, 26,678 (49.16%) were males and 27,591 (50.84%) were females. A total of 29.18% had hyperuricemia, 14.87% were vitamin A inadequate and 0.90% were vitamin A deficient. A total of 44.77% of the participants were vitamin D inadequate and 19.96% were deficient in vitamin D.

As presented in Table 2, 3244 subjects had MetS, 9391 had abdominal obesity and 8945 had high TG. Subjects with MetS, abdominal obesity and high TG had higher age, BMI, height, weight, WC, SBP, DBP, FBG, TG, serum uric acid and vitamin A, but lower HDL-C and vitamin D. Except for vitamin D, the values of the other indicators of the subjects with elevated FBG (*n* = 934) were higher than those without elevated FBG. Compared with subjects without elevated BP, subjects with elevated BP (*n* = 21,446) had higher BMI, weight, WC, SBP, DBP, FBG, TG, serum uric acid and vitamin A values, but lower age and height values. Subjects with low HDL-C (*n* = 6285) were found to have higher age, BMI, height, weight, WC, SBP, DBP, TG and serum uric acid and had lower values of HDL-C and vitamin D than those without low HDL-C. No difference was found in the values of HDL-C between the subjects with and without elevated BP, and in values of vitamin A between the subjects with and without low HDL-C. Moreover, it is noted that subjects with elevated FBG had higher HDL-C levels, and subjects with low HDL-C had lower FBG levels in the fasting venous blood tests.

### 3.2. Prevalence of MetS and Its Component Combinations among Chinese Children and Adolescents

The results regarding the prevalence of MetS and one or more of its risk factors are shown in Table 3. The prevalence of MetS among Chinese children and adolescents aged 7–17 was 5.98% and was highest in the post-pubertal stage (6.89%), in obese subjects (32.98%), and in the vitamin A sufficiency group (6.55%) and vitamin D deficiency group (7.55%). MetS was less common in females (5.66%), subjects who lived in rural areas (5.31%) and subjects without hyperuricemia (4.17%).

It was noted that the prevalence of subjects with at least one risk factor of MetS in China was up to 58.36% in this sample. A total of 21.01% had two or more risk factors and 1.19% had four or more risk factors. With age, BMI, increased vitamin A and decreased vitamin D, the prevalence of one or more of the risk factors of MetS also increased (all *p* < 0.05).

### 3.3. Prevalence of Individual MetS Components

The distributions of five individual components of MetS are displayed in Table 4. The prevalence of abdominal obesity, elevated FBG, elevated BP, high TG and low HDL-C were 17.30%, 1.72%, 39.54%, 16.48% and 11.58%, respectively. Apparently, elevated BP was most prevalent among children and adolescents aged 7–17, followed by abdominal obesity and high TG.

There were significant differences between the genders in the prevalence of abdominal obesity, elevated FBG, high TG and low HDL-C. Males had a higher prevalence of abdominal obesity, elevated FBG and low HDL-C. However, the high TG was more prevalent in females. Subjects living in urban areas had a higher prevalence of abdominal obesity and elevated FBG, but the prevalence of elevated BP was lower than that among subjects living in rural areas. We divided all subjects into three age groups, including the prepubertal, pubertal and post-pubertal stages, respectively. The prevalence of abdominal obesity, elevated FBG, high TG and low HDL-C showed an ascending trend with age, and the prevalence of elevated BP decreased with age. With increased BMI, the prevalence of all five components of MetS also increased. Among the subjects with hyperuricemia, all components were more prevalent than among subjects without hyperuricemia.

With decreased serum vitamin A, the prevalence of abdominal obesity, elevated BP, high TG and low HDL-C also decreased. There was not a statistically significant difference in the prevalence of elevated FBG between the different vitamin A categories. The prevalence of all five components of MetS showed an increasing trend with decreased serum vitamin D.

### 3.4. Associations between MetS Components and Related Factors

To further characterize the risk and protective factors of MetS and its components, a multivariate logistic regression was performed, and the corresponding results are presented in Table 5. Sex, living area, age, nutritional status, hyperuricemia, the vitamin A groups and the vitamin D groups were added to the multivariate logistic regression model. After controlling the influence of the other factors, the risk of having MetS and some of its components, such as abdominal obesity, elevated BP and high TG, was higher in females than in males (all *p* < 0.0001). However, females had a lower risk of elevated FBG (OR = 0.633, 95%CI: 0.550–0.729) and low HDL-C (OR = 0.848, 95%CI: 0.801–0.898). Compared with subjects living in urban areas, living in a rural area was positively associated with MetS, elevated BP, high TG and low HDL-C, and was negatively associated with abdominal obesity and elevated FBG. Subjects in the pubertal and post-pubertal stages had an increased risk of having MetS, abdominal obesity, elevated FBG, high TG and low HDL-C, and a decreased risk of having elevated BP. Overweight condition, obesity and hyperuricemia were positively correlated with the incidence of MetS and all five of its components.

Furthermore, we found that subjects with vitamin A inadequacy had a lower risk of having MetS, abdominal obesity, elevated BP and high TG, and had a higher risk of having low HDL-C than those with vitamin A sufficiency. Vitamin A deficiency was negatively correlated with MetS, abdominal obesity and high TG, and was positively correlated with elevated FBG and low HDL-C. Subjects with vitamin D inadequacy had a higher risk of having MetS and four of its components, excepting elevated FBG. Vitamin D deficiency was positively associated with MetS and all five of its components.

## 4. Discussion

In the present study, we reported the prevalence of MetS and its components in Chinese children and adolescents aged 7–17 using the Cook’s criteria modified for age on the basis of the NCEP-ATP III. Additionally, we explored the associations between the corresponding factors of MetS and its components using a large-scale cross-sectional study.

In our study, the prevalence of MetS was 5.98% among Chinese children and adolescents aged 7–17 from 2016–2017. Compared with the previous value of 3.37% also defined by Cook’s criteria in 2009 [27], the prevalence of MetS almost doubled over the past few years. Our results demonstrated that elevated BP (39.52%) was the most prevalent risk factor, followed by abdominal obesity and high TG, which is consistent with the study of Peige Song et al. [27]. We observed that the prevalence of MetS was higher in males (6.30%) than in females (5.66%), which conforms with previous data [34,35]. It is interesting to note that subjects with elevated FBG had higher HDL-C levels than subjects without elevated FBG, and subjects with low HDL-C had lower FBG values than those without low HDL-C. In other words, there appears to be an association between FBG and HDL-C, which acts as a kind of index of dyslipidemia. Many studies have indicated that there is a strong relationship between elevated FBG and dyslipidemia, since elevated FBG levels can lead to inflammation and oxidative stress and accelerate the progression of dyslipidemia caused by certain adverse factors, such as air pollution [36,37]. However, the association between elevated FBG and dyslipidemia is still controversial and more research on this topic is needed in the future.

We divided participants into three age-based subgroups to reflect the prepubertal, pubertal and post-pubertal stages, respectively, because the data for the Tanner stage were not available. With age, the prevalence of MetS and its components was increased. The OR of the incidence of MetS was increased among subjects who were in the pubertal (OR:1.529, 95%CI: 1.384–1.689) or post-pubertal (OR:1.596, 95%CI: 1.433–1.777) stage. We speculated that this trend could be correlated with the IR that occurs at the beginning of puberty [38]. The cause of IR in this period of life is thought to be the increase in growth hormone, sex hormone and insulin-like growth factor-1 levels [39]. However, it is remarkable to note that the subjects with elevated BP were younger than their control peers and had greater BMI values. There was a decreasing trend in the prevalence of elevated BP with increased age (*p* < 0.05), and the prevalence was up to 41.48% in the prepubertal subjects. The logistic regression indicated that age was negatively associated with the incidence of elevated BP among children and adolescents aged 7–17.

In addition, it was revealed that overweight or obese children and adolescents had a higher prevalence of MetS and its components than normal-weight subjects, which is in line with the previous studies [34,40,41]. The multivariate logistic regression also suggested that an overweight condition and obesity were positively associated with the incidence of MetS and its components, after adjusting for other factors (all *p* < 0.05). This trend could be reasonably explained by the fact that an overweight condition or obesity entails more visceral fat in the body, which has been shown to be an important trigger that can activate most of the pathways of MetS, such as chronic inflammation, hormonal activation and IR [42].

The prevalence of hyperuricemia was 29.18% among Chinese children and adolescents aged 7–17. Hyperuricemia is associated with Western diets or lifestyles [43]; thus, the spread Western diets in a non-Western country such as China has led to a high prevalence of hyperuricemia. In this study, subjects with MetS and its components had higher serum uric acid levels than their control peers. The prevalence of hyperuricemia was higher in the subjects with MetS and its components. Our study reported that the OR of the incidence of MetS was increased in subjects with hyperuricemia (OR:1.560, 95%CI: 1.430–1.703), as was the incidence of all five components of MetS (all *p* < 0.05), the values of which were similar to previous studies [9,10,27,44] but opposite to the study of Li-Li et al. [45]. The relevant pathophysiological mechanism between hyperuricemia and MetS and its component probably includes endothelial dysfunction, inflammation and/or oxidative changes induced by hyperuricemia [2,11]. In fact, serum uric acid has been recommended as a low-cost and easily accessible biochemical marker of MetS in both adults and individuals younger than 18 years old [46]. Our study seems to provide an abundance of evidence in favor of this perspective. There were also studies which concluded that vitamin D deficiency is related to hyperuricemia and that there is a bidirectional association between vitamin D and serum uric acid [47]. We could not determine a concrete relationship between them here; therefore, we included vitamin D in the model for adjustment.

Biochemical analysis of vitamin D is an accurate way of assessing vitamin status, and it can reflect the real nutritional status of vitamin D from diet intake and skin synthesis [12]. The prevalence of vitamin D insufficiency (≤20 ng/mL) was over 53% among the 15,000 children and adolescents from the 2010–2012 Chinese National Nutrition and Health Survey (CNNHS) [48]. In our study, the number was up to 64.73%. This suggests that the nutritional status of vitamin D has been continuously deteriorating, which could be reasonably attributed to poor dietary habits and decreased sun exposure [49,50]. In addition, our data indicated that the subjects with MetS and its components all had lower vitamin D levels than their control peers (all *p* < 0.05), and the prevalence of one or more risk factors increased with decreased the vitamin D, which was also the case for the five components of MetS. Moreover, our study reported that vitamin D deficiency (<12 ng/mL) was a possible risk factor of the incidence of MetS and all its components (all *p* < 0.05). Subjects with vitamin D inadequacy (12–20 ng/mL) had a high risk of having MetS, abdominal obesity, elevated BP, high TG and low HDL-C. Our results were similar to those of certain previous studies [20,51] but inconsistent with those of Chen et al. [52]. A dose–response meta-analysis reported that a 25 nmol/L increment in the serum vitamin D concentration was associated with a 20% and 15% lower risk of MetS in cross-sectional studies and cohort studies, respectively [53]. Two pathophysiologic mechanisms have been proposed to elucidate the effect of vitamin D on MetS and its components. One of these proposes that vitamin D affects insulin secretion and IR, which also play a key role in MetS development. The other is related to the potential association between obesity and vitamin D insufficiency [23]. Many clinical intervention trials have been conducted that aimed to observe the effect of vitamin D supplementation on MetS-related diseases. Rajakumar et al. [54] observed that a daily 1000 IU vitamin D supplementation had the effect of BP decrease in children after 6 months. However, it was ineffective in reducing BP in children and adolescent populations conducted by Abboud et al. [55]. It is not possible to draw a definite conclusion as to whether vitamin D insufficiency is a cause of MetS or any of its components thus far. Further investigation is warranted to determine the true role of vitamin D in the development of MetS.

Similarly, serum vitamin A levels can reflect the intake of vitamin A from diet [15]. In our study, the prevalence of vitamin A insufficiency (≤0.3 µg/mL) was 15.77%. The serum vitamin A concentration was higher in the subjects with MetS, abdominal obesity, elevated FBG, elevated BP and high TG than their control peers. There were decreasing trends in the prevalence of MetS, abdominal obesity, elevated BP, high TG and low HDL-C between the three vitamin A subgroups. Although there was no association of vitamin A inadequacy (0.2–0.3 µg/mL) with elevated FBG or vitamin A deficiency with elevated BP, we observed that vitamin A insufficiency was negatively associated with the incidence of MetS, abdominal obesity and high TG, and was positively associated with low HDL-C. Our results were similar to those of a study performed using part of our sample [56]. Other studies have also reported that vitamin A has a positive association with the incidence of MetS and some of its components [12], but a review concluded that there is no association between serum vitamin A and MetS in adults based on 11 cross-sectional studies [15]. Some mechanisms regarding the adverse effects of vitamin A on MetS and its components have been proposed. Firstly, retinol may induce oxidative stress and modulate activities of antioxidative processes, which has been observed in an in vitro study [57]. Secondly, vitamin A may be a critical regulator of β-oxidation activity, which is an important causative factor of MetS [58]. The association between vitamin A and MetS awaits further elucidation from well-designed, large-scale prospective studies.

It is clear that the adverse associations among the risk factors of MetS begin in childhood and the prevalence of MetS has been increasing; therefore, it is necessary to suggest interventions and treatments as soon as possible. Given that overweight and obese subjects had a higher risk of having MetS and all 5 components, lifestyle intervention must be the most effective measure, including dietary and physical activity. Multiple studies have shown that greater adherence to the Mediterranean diet [59] and the Dietary Approaches to Stop Hypertension (DASH) dietary pattern [60] led to a greater reversion of MetS and its components. Physical activity could improve insulin sensitivity and the endothelial vascular function [61]. Moreover, many intervention studies have shown that exercise can improve adipokine levels and has beneficial effects on the metabolism, such as the transforming growth factor-β2 (TGF-β2), an exercise-induced adipokine that improves glucose tolerance and insulin sensitivity and promotes glucose and fatty acid metabolism [62]. At the present time, there is no specific treatment for MetS in children and adolescents, other than treating the components of MetS, such as hypertension and hyperlipidemia [63].

This study had some limitations. Firstly, this study had a cross-sectional design, despite the use of a large-scale population, preventing us from forming conclusions about cause–effect relationships. A longitudinal, prospective study design is needed in the future. Secondly, we did not take into consideration dietary factors and other unmeasured potential confounding influences, such as lifestyle habits and environmental factors. In addition, the data for the insulin resistance (IR) and Tanner stage were not available for all participants in the database, and these factors are closely related with MetS. These factors should be taken into account in further studies. Nevertheless, our study still had some strengths. Using the scientific sampling method and rigorous quality control measures, a large-scale regional population was included in this research. Moreover, our study had some innovations, such as exploring the associations of serum vitamin A, vitamin D and hyperuricemia with MetS and its components among children and adolescents aged 7–17. To our knowledge, our study was first to report on the combined effects of vitamin A, vitamin D and hyperuricemia on MetS and its components among Chinese children and adolescents at the national level.

## 5. Conclusions

In summary, our study calculated the prevalence of MetS and its components in Chinese children and adolescents aged 7–17 at the national level, and we explored the correlates. The associations of living area, age, nutritional status, hyperuricemia, vitamin A and vitamin D with MetS and its components were observed after adjustment for sex. Among these factors, overweight condition, obesity and hyperuricemia had positive relationships with the incidence of MetS and all five components. Vitamin A inadequacy (0.2–0.3 µg/mL) was positively associated with low HDL-C and was negatively associated with MetS, abdominal obesity, elevated BP and high TG. Vitamin A deficiency (<0.2 µg/mL) was positively associated with low HDL-C and elevated FBG, and was negatively associated with MetS, abdominal obesity and high TG. Vitamin D inadequacy (12–20 ng/mL) was positively associated with MetS, abdominal obesity, elevated BP, high TG and low-HDL-C. Vitamin D deficiency (<12 ng/mL) had a positive relationship with MetS and all of its five components. These findings shed light on the relevant disease prevention and control methods. Further large-scale longitudinal investigations are still needed to elucidate the potential pathophysiological mechanisms and correlates of MetS in China.

## Figures and Tables

**Table 1 nutrients-14-03348-t001:** Characteristics of the basic information [*n*, %].

Variables	Total (*n* = 54,269)	Percentage (%)
Sex		
male	26,678	49.16
female	27,591	50.84
Living area		
urban	26,561	48.94
rural	27,708	51.06
Age group		
prepubertal	25,558	47.10
pubertal	15,743	29.01
post-pubertal	12,968	23.90
Nutritional status		
normal	42,258	77.87
overweight	6656	12.26
obesity	5355	9.87
Hyperuricemia		
no	38,435	70.82
yes	15,834	29.18
Vitamin A		
sufficiency	45,710	84.23
inadequacy	8071	14.87
deficiency	488	0.90
Vitamin D		
sufficiency	19,142	35.27
inadequacy	24,294	44.77
deficiency	10,833	19.96

**Table 2 nutrients-14-03348-t002:** Corresponding indicators of the subjects according to the presence of MetS and its components (median (IQR)).

Variables	All (*n* = 54,269)	With MetS (*n* = 3244)	With Abdominal Obesity (*n* = 9391)	With Elevated FBG (*n* = 934)	With Elevated BP (*n* = 21,446)	With High TG (*n* = 8945)	With Low HDL-C (*n* = 6285)
Age	11.77 (9.39, 14.19)	12.38 (10.23, 15.02) #	12.06 (9.75, 14.76) $	12.57 (10.22, 14.27) ¥	11.53 (9.18, 13.98) §	12.11 (10.09, 14.21) £	12.79 (10.36, 15.43) €
BMI	18.19 (16.19, 20.71)	24.18 (21.27, 27.31) #	23.44 (21.07, 26.14) $	19.07 (16.92, 22.15) ¥	18.73 (16.47, 21.77) §	19.66 (17.15, 23.03) £	19.71 (17.26, 23.05) €
Anthropometric measurements							
Height (cm)	150 (136, 161)	156 (146, 166) #	156 (143, 165) $	154 (142, 164) ¥	149 (135, 161) §	152 (141, 161) £	155 (142, 166) €
Weight (kg)	41.65 (30.54, 52.64)	58.40 (46.34, 73.31) #	56.60 (44.87, 69.30) $	46.30 (36.10, 57.72) ¥	42.57 (31.00, 54.80) §	46.28 (35.90, 57.20) £	48.92 (36.75, 60.00) €
WC (cm)	63.15 (56.65, 70.45)	80.00 (72.45, 88.00) #	79.30 (73.95, 85.60) $	65.20 (58.85, 74.00) ¥	64.35 (57.10, 72.95) §	67.15 (59.75, 76.40) £	68.00 (60.20, 76.50) €
SBP (mmHg)	111.33 (104.00, 119.67)	123.17 (117.00, 130.33) #	118.00 (110.00, 126.00) $	116.50 (109.00, 124.00) ¥	120.67 (114.67, 126.67) §	113.67 (105.67, 122.00) £	114.00 (105.67, 122.00) €
DBP (mmHg)	66.00 (60.67, 71.67)	71.67 (66.00, 77.00) #	68.00 (62.67, 74.00) $	68.00 (62.67, 74.33) ¥	72.33 (66.67, 77.33) §	67.00 (61.67, 73.00) £	66.33 (61.33, 72.33) €
Blood test							
FBG (mmol/L)	5.02 (4.68, 5.34)	5.07 (4.72, 5.45) #	5.09 (4.75, 5.41) $	6.29 (6.18, 6.54) ¥	5.08 (4.74, 5.39) §	5.04 (4.71, 5.36) £	4.89 (4.51, 5, 23) €
TG (mmol/L)	0.81 (0.63, 1.08)	1.52 (1.28, 1.90) #	0.98 (0.74, 1.34) $	0.90 (0.66, 1.20) ¥	0.84 (0.65, 1.13) §	1.51 (1.35, 1.81) £	1.03 (0.76, 1.43) €
HDL-C (mmol/L)	1.38 (1.18, 1.62)	1.00 (0.90, 1.20) #	1.26 (1.08, 1.47) $	1.41 (1.19, 1.67) ¥	1.38 (1.17, 1.62)	1.22 (1.04, 1.42) £	0.94 (0.85, 0.99) €
Serum uric acid (µmol/L)	308.00 (259.00, 369.00)	358 (297.0, 431.0) #	344.00 (288.00, 411.00) $	349.50 (287.36, 408.00) ¥	310.00 (260.00, 373.00) §	326.00 (274.00, 392.00) £	321.00 (261.00, 391.43) €
Vitamin A (µg/mL)	0.40 (0.33, 0.48)	0.46 (0.38, 0.55) #	0.43 (0.36, 0.52) $	0.40 (0.33, 0.50) ¥	0.40 (0.33, 0.49) §	0.45 (0.38, 0.54) £	0.40 (0.33, 0.49)
Vitamin D (ng/mL)	17.3 (12.9, 22.2)	15.50 (11.90, 20.36) #	16.4 (12.4, 21.1) $	16.55 (12.30, 21.90) ¥	16.9 (12.7, 21.9) §	16.5 (12.3, 21.4) £	15, 60 (11.50, 20.37) €

IQR: interquartile range; MetS: metabolic syndrome; FBG: fast blood glucose; BP: blood pressure; TG: triglyceride; HDL-C: high-density lipoprotein cholesterol; WC: waist circumference; BMI: body mass index; SBP: systolic blood pressure; DBP: diastolic blood pressure; # *p* < 0.05, subjects with MetS vs. without MetS; $ *p* < 0.05, subjects with abdominal obesity vs. without abdominal obesity; ¥ *p* < 0.05. subjects with elevated FBG without elevated FBG; § *p* < 0.05, subjects with elevated BP vs. without elevated BP; £ *p* < 0.05, subjects with high TG vs. without high TG; € *p* < 0.05, subjects with low HDL-C vs. without low HDL-C.

**Table 3 nutrients-14-03348-t003:** Prevalence of MetS and one or more risk factors of MetS [*n* (%)].

Variables	Number of Risk Factors			
	≥1	≥2	≥3 (MetS)	≥4
All	31,671 (58.36)	11,417 (21.04)	3244 (5.98)	648 (1.19)
Sex	a		a	a
male	15,367 (57.60)	5582 (20.92)	1681 (6.30)	374 (1.40)
female	16,304 (59.09)	5835 (21.15)	1563 (5.66)	274 (0.99)
Living area		a	a	a
urban	15,529 (58.47)	5933 (22.34)	1771 (6.68)	369 (1.39)
rural	16,142 (58.26)	5484 (19.79)	1471 (5.31)	279 (1.01)
Age group	b	b	b	b
prepubertal	14,641 (57.29)	4797 (18.77)	1270 (4.97)	201 (0.79)
pubertal	9441 (59.97)	3685 (23.41)	1080 (6.86)	238 (1.51)
post-pubertal	7589 (58.52)	2935 (22.63)	894 (6.89)	209 (1.61)
Nutritional status	b	b	b	b
normal	21,284 (50.37)	4832 (11.43)	685 (1.62)	45 (0.11)
overweight	5272 (79.21)	2646 (39.75)	793 (11.91)	151 (2.27)
obesity	5115 (95.52)	3939 (73.56)	1766 (32.98)	452 (8.44)
Hyperuricemia	a	a	a	a
no	21,578 (56.14)	6787 (17.66)	1602 (4.17)	246 (0.64)
yes	10,093 (63.74)	4630 (29.24)	1642 (10.37)	402 (2.54)
Vitamin A	b	b	b	b
sufficiency	27,083 (59.25)	10,234 (22.39)	2994 (6.55)	613 (1.34)
inadequacy	4330 (53.65)	1117 (13.84)	241 (2.99)	35 (0.43)
deficiency	258 (52.87)	66 (13.52)	9 (1.84)	0 (0.00)
Vitamin D	b	b	b	b
sufficiency	10,385 (54.25)	3306 (17.27)	852 (4.45)	154 (0.80)
inadequacy	14,451 (59.48)	5345 (22.00)	1574 (6.48)	321 (1.32)
deficiency	6835 (63.09)	2766 (25.53)	818 (7.55)	173 (1.60)

MetS: metabolic syndrome; a, Chi-square test, *p* < 0.05; b, Cochran–Armitage trend test, *p* < 0.05.

**Table 4 nutrients-14-03348-t004:** Prevalence of the components of MetS in the research population [*n* (%)].

Variables	Abdominal Obesity	Elevated FBG	Elevated BP	High TG	Low HDL-C
All	9391 (17.30)	934 (1.72)	21,446 (39.52)	8945 (16.48)	6285 (11.58)
Sex	a	a		a	a
male	4752 (17.81)	589 (2.21)	10,529 (39.47)	3864 (14.48)	3286 (12.32)
female	4639 (16.81)	345 (1.25)	10,917 (39.57)	5081 (18.42)	2999 (10.87)
Living area	a	a	a		
urban	5600 (21.08)	552 (2.08)	10,117 (38.09)	4308 (16.22)	3043 (11.46)
rural	3791 (13.68)	382 (1.38)	11,329 (40.89)	4637 (16.74)	3242 (11.70)
Age group	b	b	b	b	b
prepubertal	4130 (16.16)	358 (1.40)	10,602 (41.48)	3617 (14.15)	2207 (8.64)
pubertal	2747 (17.45)	357 (2.27)	6102 (38.76)	3122 (19.83)	2123 (13.49)
post-pubertal	2514 (19.39)	219 (1.69)	4742 (36.57)	2206 (17.01)	1955 (15.08)
Nutritional status	b	b	b	b	b
normal	1791 (4.24)	634 (1.50)	14,738 (34.88)	5572 (13.19)	4112 (9.73)
overweight	2970 (44.62)	147 (2.21)	3291 (49.44)	1454 (21.84)	1006 (15.11)
obesity	4630 (86.46)	153 (2.86)	3417 (63.81)	1919 (35.84)	1167 (21.79)
Hyperuricemia	a	a	a	a	a
no	5163 (13.43)	498 (1.30)	14,916 (38.81)	5633 (14.66)	4010 (10.43)
yes	4228 (26.70)	436 (2.75)	6530 (41.24)	3312 (20.92)	2275 (14.37)
Vitamin A	b		b	b	b
sufficiency	8495 (18.58)	787 (1.72)	18,198 (39.81)	8278 (18.11)	5185 (11.34)
inadequacy	854 (10.58)	134 (1.66)	3071 (38.05)	635 (7.87)	1031 (12.77)
deficiency	42 (8.61)	13 (2.66)	177 (36.27)	32 (6.56)	69 (14.14)
Vitamin D	b	b	b	b	b
sufficiency	2801 (14.63)	300 (1.57)	7147 (37.34)	2793 (14.59)	1661 (8.68)
inadequacy	4464 (18.37)	420 (1.73)	9796 (40.32)	4111 (16.92)	2909 (11.97)
deficiency	2126 (19.63)	214 (1.98)	4503 (41.57)	2041 (18.84)	1715 (15.83)

FBG: fast blood glucose; BP: blood pressure; TG: triglyceride; HDL-C: high-density lipoprotein cholesterol; a, Chi-square test, *p* < 0.05; b, Cochran–Armitage trend test, *p* < 0.05.

**Table 5 nutrients-14-03348-t005:** Multivariable logistic regression analysis of MetS and its components with related factors.

Variables	MetS		Abdominal Obesity		Elevated FBG		Elevated BP		High TG		Low HDL-C	
	OR (95%CI)	*p*	OR (95%CI)	*p*	OR (95%CI)	*p*	OR (95%CI)	*p*	OR (95%CI)	*p*	OR (95%CI)	*p*
Living area												
urban	ref		ref		ref		ref		ref		ref	
rural	1.171 (1.080, 1.268)	0.0001	0.787 (0.738, 0.839)	<0.0001	0.694 (0.607, 0.793)	<0.0001	1.249 (1.206, 1.295)	<0.0001	1.223 (1.167, 1.283)	<0.0001	1.106 (1.048, 1.167)	0.0003
Age group												
prepubertal	ref		ref		ref		ref		ref		ref	
pubertal	1.529 (1.384, 1.689)	<0.0001	1.519 (1.398, 1.650)	<0.0001	1.485 (1.265, 1.743)	<0.0001	0.875 (0.837, 0.914)	<0.0001	1.329 (1.255, 1.408)	<0.0001	1.668 (1.558, 1.786)	<0.0001
post-pubertal	1.596 (1.433, 1.777)	<0.0001	2.048 (1.878, 2.233)	<0.0001	1.078 (0.896, 1.298)	0.4249	0.794 (0.756, 0.833)	<0.0001	1.055 (0.989, 1.124)	0.1019	1.943 (1.807, 2.088)	<0.0001
Nutritional status												
normal	ref		ref		ref		ref		ref		ref	
overweight	8.049 (7.227, 8.966)	<0.0001	20.420 (18.997, 21.950)	<0.0001	1.277 (1.062, 1.535)	0.0093	1.866 (1.770, 1.967)	<0.0001	1.859 (1.740, 1.987)	<0.0001	1.654 (1.533, 1.785)	<0.0001
obesity	31.905 (28.839, 35.296)	<0.0001	200.284 (180.901, 221.743)	<0.0001	1.554 (1.290, 1.872)	<0.0001	3.344 (3.146, 3.554)	<0.0001	3.839 (3.593, 4.103)	<0.0001	2.826 (2.618, 3.051)	<0.0001
Hyperuricemia												
no	ref		ref		ref		ref		ref		ref	
yes	1.560 (1.430, 1.703)	<0.0001	1.610 (1.498, 1.731)	<0.0001	1.739 (1.501, 2.015)	<0.0001	1.049 (1.005, 1.094)	0.0293	1.326 (1.257, 1.400)	<0.0001	1.084 (1.018, 1.154)	0.0117
Vitamin A												
sufficiency	ref		ref		ref		ref		ref		ref	
inadequacy	0.647 (0.560, 0.748)	<0.0001	0.745 (0.671, 0.827)	<0.0001	1.186 (0.980, 1.435)	0.0803	0.931 (0.885, 0.980)	0.0059	0.432 (0.396, 0.471)	<0.0001	1.378 (1.279, 1.486)	<0.0001
deficiency	0.359 (0.180, 0.716)	0.0036	0.413 (0.267, 0.639)	<0.0001	1.917 (1.096, 3.356)	0.0226	0.853 (0.705, 1.030)	0.0988	0.357 (0.248, 0.514)	<0.0001	1.570 (1.209, 2.038)	0.0007
Vitamin D												
sufficiency	ref		ref		ref		ref		ref		ref	
inadequacy	1.364 (1.240, 1.500)	<0.0001	1.270 (1.178, 1.369)	<0.0001	1.102 (0.947, 1.283)	0.2094	1.146 (1.100, 1.193)	<0.0001	1.120 (1.060, 1.183)	<0.0001	1.315 (1.232, 1.404)	<0.0001
deficiency	1.646 (1.468, 1.845)	<0.0001	1.428 (1.302, 1.565)	<0.0001	1.283 (1.067, 1.543)	0.0081	1.256 (1.193, 1.322)	<0.0001	1.269 (1.186, 1.357)	<0.0001	1.712 (1.587, 1.848)	<0.0001

The multivariate logistic regression model was adjusted according to sex. MetS: metabolic syndrome, FBG: fast blood glucose; BP: blood pressure; TG: triglyceride; HDL-C: high-density lipoprotein cholesterol, OR: odds ratio, CI: confidence interval.

## Data Availability

According to the policy of National Institute for Nutrition and Health, Chinese Center for Disease Control and Prevention, data related in this research are not allowed to be disclosed.

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
