# Peer review of "Prevalence and Correlates of Metabolic Syndrome and Its Components in Chinese Children and Adolescents Aged 7–17: The China National Nutrition and Health Survey of Children and Lactating Mothers from 2016–2017"

_nutrients, 2022, doi:10.3390/nu14163348_

Round 1

Reviewer 1 Report

Dear Authors,

The study is well conducted and the sample size is very large, so appropriate conclusions can be drawn.

Minor improvements can be made, as follows:

The section results

Line 168: should be mentioned only Table 1. The explanations for table 2 are below. In our opinion weight and height,  as individual indicators are not relevant, only the BMI is a relevant indicator.

Line 220 : What do you mean with "but it is opposite in subjects with elevate BP"

Discussion

Line 272 -277. The explanation is unclear. Please rewrite

Line 361 In our opinion it is better to write retinol induces oxidative stress rather than vitamin A because carotenes the pro-viatmin A is a well-known antioxidant (you also wrote this in the introduction section).

Line 374 Please rewrite the sentence. How can improve the adipokine level?

Conclusions

I recommend rewriting the conclusion more clearly, according to the results of the study (for example specify the levels of vitamins).

Kind regards

Author Response

Dear reviewer,

Thank you for your letter and comments concerning our manuscript entitled ” Prevalence and Correlates of Metabolic Syndrome and its components in Chinese children and adolescents aged 7-17: The China National Nutrition and Health Surveillance of Children and Lactating Mothers in 2016-2017” (ID: nutrients-1814155). Based on your suggestions, we have accordingly revised our manuscript. Below you could find the point-to-point response to the questions regarding the manuscript.

We hope that our answers have satisfied your comments and look forward to your response.

Warm regards,

Reviewer 2 Report

Interesting paper on the prevalence and correlates of metabolic syndrome and its components in Chinese children and adolescents aged 7 to 17 years.

Well-designed study, with methodology appropriate to the objectives outlined. Statistical study appropriate to the data obtained. Discussion based on the data collected and based on the referred bibliography.

Author Response

Dear reviewer,

Thank you for your letter and comments concerning our manuscript entitled ” Prevalence and Correlates of Metabolic Syndrome and its components in Chinese children and adolescents aged 7-17: The China National Nutrition and Health Surveillance of Children and Lactating Mothers in 2016-2017” (ID: nutrients-1814155). It’s very glad to have your confirmation to our work. We believe it is meaningful to improve the health of children and adolescents. Thanks again to your hard work.

Warm regards